ecology, health and disease and epidemiology

cross-species transmission, zoonotic disease risk, machine learning, one health, big data, pathogen spillover

**Author for correspondence:**
Maya Wardeh
e-mail: maya.wardeh@liverpool.ac.uk

# Integration of shared-pathogen networks and machine learning reveals the key aspects of zoonoses and predicts mammalian reservoirs

Maya Wardeh[1], Kieran J. Sharkey[2] and Matthew Baylis[3,4]

[1]Department of Epidemiology and Population Health, Institute of Infection and Global Health, University of Liverpool, Liverpool Science Park IC2 Building, 146 Brownlow Hill, Liverpool L3 5RF, UK
[2]Department of Mathematical Sciences, University of Liverpool, Peach Street, Liverpool L69 7ZL, UK
[3]Department of Epidemiology and Population Health, Institute of Infection and Global Health, University of Liverpool, Leahurst Campus, Chester High Road, Neston CH64 7TE, UK
[4]Health Protection Research Unit in Emerging and Zoonotic Infections, University of Liverpool, Liverpool L69 7BE, UK

MW, 0000-0002-2316-5460; MB, 0000-0003-0335-187X

Diseases that spread to humans from animals, zoonoses, pose major threats to human health. Identifying animal reservoirs of zoonoses and predicting future outbreaks are increasingly important to human health and well-being and economic stability, particularly where research and resources are limited. Here, we integrate complex networks and machine learning approaches to develop a new approach to identifying reservoirs. An exhaustive dataset of mammal–pathogen interactions was transformed into networks where hosts are linked via their shared pathogens. We present a methodology for identifying important and influential hosts in these networks. Ensemble models linking network characteristics with phylogeny and life-history traits are then employed to predict those key hosts and quantify the roles they undertake in pathogen transmission. Our models reveal drivers explaining host importance and demonstrate how these drivers vary by pathogen taxa. Host importance is further integrated into ensemble models to predict reservoirs of zoonoses of various pathogen taxa and quantify the extent of pathogen sharing between humans and mammals. We establish predictors of reservoirs of zoonoses, showcasing host influence to be a key factor in determining these reservoirs. Finally, we provide new insight into the determinants of zoonosis-sharing, and contrast these determinants across major pathogen taxa.

## 1. Introduction

Recent years have seen significant outbreaks of several emerging zoonotic diseases, ranging from the well known (e.g. Ebola), to the previously little known (Zika), to the novel (Middle East respiratory syndrome, MERS). It is well established that most communicable human diseases have animal origins [1,2], and over two-thirds of emerging human pathogens originate from mammals [3,4]. In addition, endemic zoonoses continue to pose major threats to human health, well-being [5] and economic security [6]. Despite the importance of cross-species spillover transmission of zoonotic pathogens [7], there has been relatively little attention paid to how these pathogens are shared between humans and non-human animals. Indeed, in many cases, the animal sources of major human outbreaks have only been identified after the outbreaks have occurred. There is a pressing need to increase our understanding of how pathogens are shared between humans and animals, so that we can anticipate possible outbreaks in advance [8,9].

Efforts have been made to address this issue by attempting to find patterns in the distribution of pathogens among mammals, or incorporating various

analytical techniques to explain sharing of these pathogens with humans. Many of these studies have focused on factors promoting pathogen sharing within specific taxonomic host groups (notably primates [9,10], bats [11,12], carnivores [13] and rodents [11,12,14]), non-taxonomic groups (e.g. domestic animals [15]) or have limited their scope to certain pathogens, or taxa of pathogens, particularly viruses [16]. Here, we present a comprehensive species-level analysis of pathogen sharing between all known non-human mammalian hosts and humans, contrasting the factors promoting sharing of various taxa of pathogens (e.g. bacteria versus viruses).

Networks of shared pathogens have been gaining popularity as useful tools to investigate pathogen sharing and transmission [9,12,15,17,18]. Here, we integrate networks of shared pathogens with predictive machine learning to answer three key questions in relation to the link between sharing of pathogens between mammals and mammalian reservoirs of zoonoses. Can we identify important host species and quantify their roles in sharing and transmitting different pathogen taxa? Can these roles be integrated with mammalian traits to predict reservoirs of zoonoses? And which host traits and roles best explain the number of pathogens shared between mammalian hosts and humans?

## 2. Data and methods

### (a) Host–pathogen species interactions and network formulation

We extracted interactions between non-human mammals and their pathogens from the enhanced infectious diseases database (EID2) [19]. These interactions formed a bipartite network with nodes represented hosts and pathogens, and links indicated whether pathogens have been found in hosts. We further checked these interactions to ascertain whether the putative pathogens caused a disease or an opportunistic infection in at least one species of mammal (including humans). This resulted in 16 548 species-level host–pathogen interactions between 3986 pathogen species (bacteria = 885, fungi = 251, helminth = 1000, protozoa = 404 and virus = 1446) and 1560 non-human mammalian species.

We projected this bipartite network into a unipartite network where nodes represented host species and edges quantified shared pathogens. The motivation behind this projection is twofold: (i) it enabled us to investigate dynamics of pathogen sharing via ecological network analysis tools [9,11], and (ii) it facilitated identification of important non-human mammalian species via centrality measures. In addition to the network encompassing all pathogen taxa (including fungi), we generated eight additional networks: bacteria (including Gram variable), Gram+ bacteria, Gram− bacteria, helminths, protozoa, viruses (including retro-transcribing), DNA viruses and RNA viruses. Prions were not included in this study (electronic supplementary material, note S1 lists further information).

We computed various network statistics to contrast key aspects of pathogen sharing across our selected taxa (table 1). These included: *transitivity* (if two nodes are connected, the probability that their neighbours are also connected); *density* (the proportion of potential connections in a network that are actual connections); and *modularity* (the number of edges falling within groups minus the

expected number in an equivalent network with edges placed at random [20]). We also calculated *network-level E–I index* based on species orders. Given a categorical node attribute describing mutually exclusive groups (in our case, order), the E–I index represents a ratio of external (with other orders) to internal (within order) edges. A positive network-level E–I index indicates a tendency of hosts to share pathogens with species outside their order (i.e. extrovert), whereas a negative E–I index indicates a tendency to share pathogens within orders (i.e. introvert).

### (b) Centrality measures in networks of shared pathogens

Various metrics of node centrality have been explored as proxies to host importance in networks of shared pathogens [9,12,17,21]. Each of these metrics reflects distinct characteristics of the roles host species play in shared-pathogen networks and tends to fall into wider categories including: degree and eigenvalue-derived measures, capturing direct sharing of pathogens among hosts; and distance-based measures, relating to indirect sharing of pathogens. However, due to the large number of available metrics in each category, determining which ones are best suited for identifying important hosts in networks is not straightforward. To address this, we use principal component analysis (PCA) to determine the efficacy of a wide range of well-established centrality measures and subsequently select the ones best suited to our networks [22].

To achieve this, we computed seven degree- [23–25] and eigenvalue-derived [28–30] centrality measures, and six distance-based measures [24,29,30] in each of our networks. Electronic supplementary material, note S2 lists details and definitions of selected measures. We then performed PCA analyses using the R package *FactoMineR* [31] to determine the influence of our selected measures within each network. We calculated the contributions of each centrality measure to the first and second principal components (explained 89.6% of variance on average; first = 76.3%, second = 13.3%) to determine those which explained most of the variation in these components.

Following this, Opsahl degree centrality (ODC) (mean contribution to first = 8.86% and second = 0.34%) and Opsahl betweenness centrality (OBC) (3.14%, 24.18%) emerged as the two measures with the greatest contribution on average to the first and second component, respectively, across our networks, and were therefore selected in our analyses. ODC can be thought of as a quantifier of the host reachability within the network, whereas OBC reflects the ability of the host to bridge various communities. Hosts with high OBC receive/transmit different types of pathogens from these communities without necessarily communicating these pathogens across communities. Electronic supplementary material, note S2 provides details of included measures and the PCA analyses.

### (c) Novel metric of node influence: indirect influence

Unipartite projection of host–pathogen bipartite networks results in inevitable loss of information [32]. Let us assume, for instance, that we have three mammalian species: A, B and C, and that A and B share 10 protozoan agents, whereas B and C share 5 protozoan agents. Here, we have two

**Table 1.** Network statistics. Zoonoses here refer to the number of pathogens found in at least one mammalian host species and humans in each network. Fields in italic represent values unchanged in the permutation tests (details of permutation tests are listed in electronic supplementary material, note S1). Modularity ranges from −1 to +1, with higher values indicating strong community structure within the network. Order E–I index also ranges from −1 (fully introvert) to +1 (fully extrovert).

| | all | bacteria | Gram− | Gram+ | helminth | protozoa | virus | DNA | RNA |
|---|---|---|---|---|---|---|---|---|---|
| hosts | *1560* | *722* | *307* | *660* | *706* | *749* | *1001* | *451* | *836* |
| zoonoses reservoirs | *1242* | *680* | *295* | *615\** | *406\** | *659\*\** | *672* | *97\*\** | *588* |
| pathogens | *3986* | *885* | *344* | *539* | *1000* | *404* | *1446* | *769* | *616* |
| zoonoses | *1010* | *489* | *212* | *276* | *118* | *68* | *185* | *43* | *130* |
| pathogens /host | 10.96** | 5.99** | 4.47** | 4.40** | 5.58** | 3.55** | 4.77** | 4.05** | 3.30** |
| shared pathogens/ host | 9.55** | 5.38** | 3.92** | 3.99** | 4.96** | 3.33** | 3.77** | 2.78** | 2.87** |
| zoonoses /host | 5.93** | 4.8** | 3.8** | 3.41** | 1.88** | 2.05** | 1.9** | 0.64** | 1.73** |
| host–pathogen interactions | *16 548* | *4328* | *2906* | *1373* | *3940* | *2659* | *4775* | *1599* | *2760* |
| edges | 162 694** | 56 809** | 11 239** | 51 418** | 14 889* | 94 651** | 41 086** | *5500* | 35 279** |
| transitivity | 0.64** | 0.66** | 0.66** | 0.69** | 0.50** | 0.86** | 0.67** | 0.74** | 0.71** |
| density | 0.14** | 0.22** | 0.24** | 0.24** | 0.06* | 0.34** | 0.08** | 0.09** | 0.10** |
| modularity | 0.15** | *0.15* | *0.15* | *0.18* | 0.43** | *0.13* | 0.36** | 0.62** | *0.27* |
| order E–I index | 0.5** | 0.52** | 0.49** | 0.52** | −0.09** | 0.64** | 0.04** | −0.73** | 0.17** |
| mean degree | 216** | 157** | 73** | 156** | 42* | 253** | 82** | 31** | 84** |
| mean ODC | 282** | 192** | 86** | 181** | *53* | 276** | 96** | 36** | 93** |
| mean OBC | 979** | *338* | *124* | *290* | 505** | 258** | 560** | *185* | *446* |
| mean II | 0.50** | 0.58** | 0.57** | 0.61** | 0.37** | 0.67** | 0.44** | 0.44** | 0.48** |

*Permutation *p*-value ≤ 0.05; **\*\****p*-value ≤ 0.01.

extreme scenarios, as follows. (i) The protozoans are all different; A and C do not share any protozoan pathogens, which in turn means that there is no flow of protozoan pathogens from A to C despite a path existing between the two (via B). (ii) The five protozoans shared by B and C are among the 10 shared by A and B and, therefore, half of A's protozoans are shared with C, and all of C's are shared with A. Both scenarios have implications on the traditional centralities analysed in the previous subsection. To address these issues, we developed a novel entropy-based metric, which we term indirect influence (II). II captures the influence each host species exercises within the unipartite network by measuring the number and frequency of pathogens this host spreads indirectly through its neighbouring species if it were to interact with each of them in isolation to the remainder of the network. Entropy enables us to capture uncertainties in the destination of pathogens as a function of the original host [33] at three levels: (i) a species which shares a few pathogens with many neighbours: entropy captures uncertainty in which neighbour it could influence indirectly; (ii) a species which shares many pathogens with few neighbours: entropy allows us to capture uncertainty in pathogens shared; and (iii) a species which shares many pathogens with many neighbours will have high centrality due to uncertainties in both pathogens shared and neighbours influenced. In addition, using entropy allows us to avoid assessing the many paths connecting all node pairs and instead, focus on the potential of the node (i.e. the host species) to diversify pathogen propagation [33,34].

II uses the above while taking into account the fact that in shared-pathogen networks, nodes share pathogens with their immediate neighbours only, which means that the distance between two not-directly connected nodes equals infinity regardless of whether a path exists between the two nodes or not. Formally, for each node $i \in N$, connected to $n_i$ neighbours with cardinality $|n_i|$, indirect influence of node $i$ ($\mathrm{II}_i$) is as follows:

$$\mathrm{II}_i = \mathrm{II}_i^d + \mathrm{II}_i^p + \mathrm{II}_i^f, \tag{2.1}$$

where $\mathrm{II}_i^d$, $\mathrm{II}_i^p$ and $\mathrm{II}_i^f$ are the node's indirect degree influence, indirect pathogen influence and indirect frequency influence, respectively (equations (2.2)–(2.4)).

$$\mathrm{II}_i^d = -\sum_{j \in N} \frac{|n_{ij}|}{|n_i|} \times \log \frac{|n_{ij}|}{|n_i|}, \tag{2.2}$$

where $n_{ij}$ is the neighbours of $i$ reachable through $j$, $n_{ij}$, $|n_{ij}|$ is the cardinality of $n_{ij}$.

$$\mathrm{II}_i^p = -\sum_{j \in N} \frac{|p_{ij}|}{|p_i|} \times \log \frac{|p_{ij}|}{|p_i|}, \tag{2.3}$$

where $p_{ij} = p_i \cap p_j$; $p_i, p_j$ is the pathogens nodes $i$, $j$ share with their neighbours.

$$\mathrm{II}_i^f = -\sum_{j \in N} \frac{f_{ij}}{f_{ij}} \times \log \frac{f_{ij}}{f_{ij}}, \tag{2.4}$$

where $f_{ij}$ is the frequency of sharing from $i$ through $j$, and $f_i$ is the frequency by which $i$ shares pathogens with all its neighbours.

Measuring II of a host is essential to capturing the intricacies of pathogen sharing not covered by the conventional measures analysed above, namely the range and speed by which host species propagate their pathogens to their neighbours. In this, our measure is similar to other entropy-based measures [34], as well as to established influence measures such as Katz centrality [28,29], but it focuses on the species influence in its neighbour space rather than across the whole network. To further assess the relationship between II and established centrality metrics, we performed a correlation analysis of centrality measures in our networks which revealed that II correlated (on average across all networks) with closeness [29] (0.88) and Opsahl closeness [24] (0.82) centralities, and showed least correlation with OBC (0.16). Further details are listed in electronic supplementary material, note S2.

## (d) Predictors
### (i) Predictors of centrality and influence in networks of shared pathogens

We compiled a set of predictors of importance of host species and roles they play in networks of shared pathogens as quantified by the centrality and influence measures discussed above as follows.

*Host orders and domestication status:* we extracted host orders from EID2 [19]. Orders have been found to affect sharing of pathogens among species [16] and therefore their position in our networks. We classified hosts into three non-taxonomic groups: domesticated, semi-domesticated and wild (electronic supplementary material, note S1 expands these definitions).

*Research effort:* we quantified research effort for each mammalian species to be the total count of the host (and any of its subspecies) genetic sequences and publications as indexed by EID2 [19]. This enabled us to control for research effort by including it as an independent variable in our models.

*Pathogen diversity:* we used two metrics of pathogen taxonomic specificity, namely: (i) taxonomic distinctness specificity index $S_{TD}$ [35,36] and (ii) variance in taxonomic distinctiveness specificity index $VarS_{TD}$ [35]. We calculated these metrics based on NCBI taxonomy as used by EID2 [19]. A higher value of $S_{TD}$ indicated that on average the host has been exposed to pathogens that are not closely related. $VarS_{TD}$ captured higher-level asymmetries that might have been missed by the former metric.

*Species traits:* we compiled species traits from online databases and literature [16,37–40]. We included body mass and body length as they have been shown to affect the number of pathogens harboured by a host, or shared with other hosts, as well as acting as proxies, together with maximum longevity, indicative of host metabolism and adaptation to environment. We included reproductive traits (litters per year, litter size, weaning age, gestation period and age at sexual maturity) which could be viewed as proxies to within-host–pathogen dynamics and therefore may influence the pathogens harboured by the host and the host position within the network. We also calculated the proportional use of 10 diet categories as presented in EltonTraits [39]. We used these categories as independent variables to assess the effect of variations in each category on the position of hosts in our networks.

*Geography and habitat:* we included species geographical area range [40] (in $km^2$) as we assumed that hosts with wider range might come into contact with larger number of host species, than those confined to smaller areas, and may

therefore have higher centrality values. We incorporated habitat usage [40] as multiple binary indicators of whether a species uses one or more of 14 natural and artificial habitats. We hypothesized that habitat usage affects both the pathogens and hosts with which mammals come into contact and therefore influences centrality measures.

*Phylogeny:* we calculated pairwise phylogenetic distances between mammalian host species in each network using a recent mammalian supertree [41]. We used these distances to compute two hybrid network-phylogeny measures for each host: neighbours distinctness specificity index ($S_{ND}$) and variance in neighbours distinctiveness specificity index (VarS$_{ND}$). We based those measures on $S_{TD}$ [34–36] and VarS$_{TD}$ [35,37]. Higher values of these measures indicated the hosts share pathogens with distant or varied mammalian species, whereas lower values indicated that sharing of pathogens is more localised, to genus or family level. Formally, we defined our measures as follows. For each host node in the network $i$ with a set of neighbours $n_i$,

$$S_{ND i} = 2 \times \frac{\Sigma\Sigma_{i<j}\, \omega_{jk}}{|n_i|(|n_i| - 1)} \,, \tag{2.5}$$

$j, \underline{k} \in \underline{n_i}$; $\omega_{ij}$ is the phylogenetic distance between $j$ and $k$,

$$\text{VarS}_{ND h} = \frac{\Sigma\Sigma_{i<j}\, (\omega_{ij} - \bar{\omega})^2}{|n_i|(|n_i| - 1)}, \ \bar{\omega} = S_{ND h}. \tag{2.6}$$

In addition, we calculated the evolutionary distinctiveness of each host species in our tree using fair proportions [42], as implemented in the R package *picante* [43].

### (ii) Predictors of reservoir of zoonoses and number of zoonoses shared with mammalian hosts

We incorporated taxonomic orders, domestication status and geographical range of species described above in our models to predict reservoirs of zoonoses and predict the number of zoonoses shared with mammalian hosts. To account for research biases in zoonotic pathogens (which tend to be more studied in general), we supplemented host research effort by computing Shannon entropy of research effort (sum of publications and sequences) of pathogens of each host. Shannon entropy accounted for the proportion of research on each pathogen species. Larger values of Shannon entropy indicate larger research effort and a more even distribution of this effort among different pathogen species in the mammalian hosts.

*Position in network:* we included the centrality measures discussed above: ODC, OBC and II as predictors in our zoonoses analyses. Each of these measures reflects a unique characteristic of the role hosts play in the network and therefore a different aspect of sharing of pathogens among mammals.

*Relation to human:* In addition to their domestication status, we categorized the nature of each mammal interaction with humans into four binary indicators [40]: food source, companion/pet, other usages (e.g. clothes, transportation) and hunted (for sport or food).

*Distance to human:* We used the mammalian supertree [41] from the previous subsection to compute the phylogenetic distance between each host species and humans. In addition, we quantified three separate ecological distances between

these hosts and humans: (i) life traits distance (using all life traits listed above), (ii) habitat distance (including habitat binary indicators from the previous section), and (iii) diet distance (including the 10 proportional categories used above). We based these three distances on a generalized form of Gower's distance matrices [21,44,45].

### (e) Ensembles construction

We developed ensembles to investigate if our chosen centrality measures can be explained by life traits of hosts, their phylogeny and relation to their neighbours in networks. Each of our ensembles comprised six learners: stochastic gradient boosting/boosted regression trees (gbm), support vector machines, elastic generalized linear models, nearest neighbours, decision trees and random forests. We tuned and trained these learners using the R packages *Caret* [31,32] and *caretEnsemble* [33] for nine categories of pathogens corresponding to the nine networks we constructed above. We constructed our ensembles from these learners using linear greedy optimization (as implemented in *caretEnsemble* [33]) to minimize root mean square error (RMSE). We validated our ensembles and their constituents using 10-fold cross-validation. We repeated this process 100 times, to attend to uncertainties in the cross-validation processes, and to generate empirical confidence intervals. Predictions were generated using the median values of these repeats.

The relative contribution (importance) of predictors included in our ensembles (as discussed above) was computed by averaging the predictive power of variables across the base models with weights equal to the contribution of these models to the ensemble (as implemented in the R package *caretEnsemble* [33]). This enabled us to assign a value between 0 and 100 to each predictor, with larger values indicating a larger relative influence on the ensemble.

Using similar methodology to the one outlined above, we constructed ensemble models to answer two questions: (i) which mammals are more likely to harbour zoonotic pathogens? (ii) Can the number of zoonoses be explained by centrality and/or host traits? Our ensembles to answer the first question (a classification problem) were optimized to maximize the area under the ROC curve (AUC). Our ensembles to answer the second question (regression problem) were constructed, tuned and assessed similarly to the above subsection.

We assessed the performance of our ensembles and their constituent learners via a comprehensive set of performance metrics (as detailed in electronic supplementary material, note S3).

## 3. Results

### (a) Description of networks

Network statistics varied across the pathogen taxa studied, as illustrated in table 1. Viruses (all, DNA and RNA) and helminth networks were less dense than bacteria and protozoa networks (electronic supplementary material, figure S1). The protozoa network exhibited the highest level of transitivity, indicating higher probability that if two host species shared protozoan pathogens, their neighbours were also connected. Network-level order E–I index revealed that hosts of helminth and DNA viruses (and to lesser extent all viruses

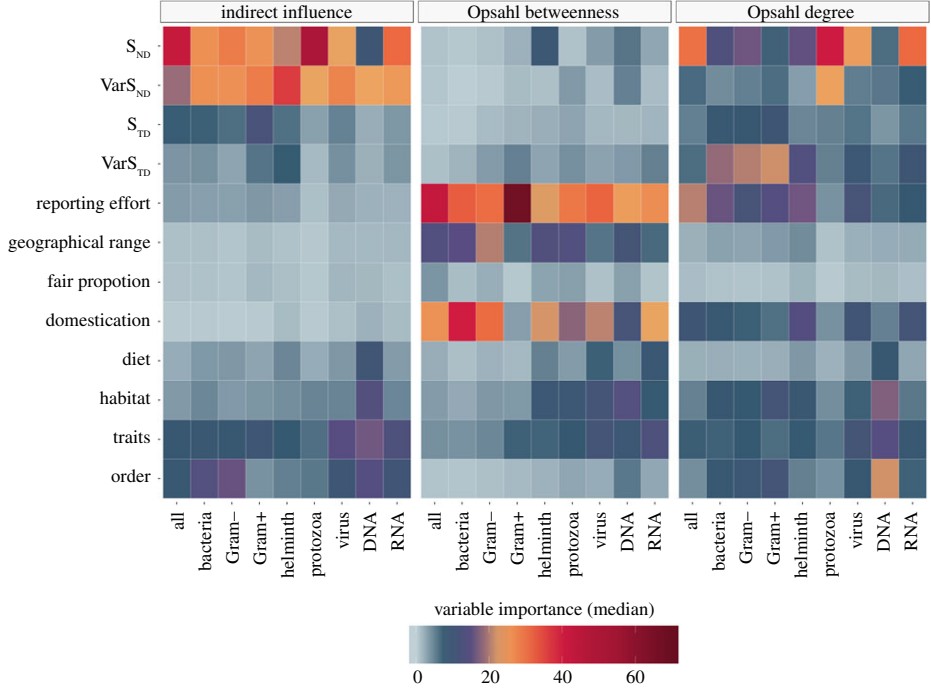

**Figure 1.** The relative influence of predictors included in our ensemble models to explain centrality measures in networks of shared pathogens among non-human mammals. For the purposes of this figure, the contribution of individual predictors in the categories of life traits, habitat, host order and diet was summed. Details of individual contributions of all predictors over the 100 runs of each model are presented in electronic supplementary material, note S4. (Online version in colour.)

and RNA viruses) tended to share pathogens in these taxa with members of their orders (table 1).

## (b) Ensemble models to explain centrality and influence in networks of shared pathogens

### (i) Predictors of centrality

We calculated the relative contribution (importance) of each variable to the ensembles by averaging its predictive power of base learners with weights equal to their contribution to the greedy ensemble. The influence of predictors varied per centrality measure and pathogen type, as illustrated in figure 1 (and further explored in electronic supplementary material, note S4).

Neighbours' phylogenetic specificity ($S_{ND}$) was the top predictor overall of II (median = 26.4%; 95% CIs [9.24%, 51.1%]) and ODC (16.7%; [5.89%, 41.0%]). However, there were variations across networks (figure 1). ODC in the network of DNA viruses was best explained by species order, particularly, Carnivora (median = 11.3%; 95% CIs [10.5%, 12.7%]). ODC in networks of bacterial agents was best explained by pathogen diversity $VarS_{TD}$ as follows: all bacteria (median = 18.4%; 95% CIs [17.3%, 19.7%]), Gram+ (21.7%; [20.7%, 23.1%]), Gram− (20.2%; 95% CIs [18.9%, 21.4%]).

Research effort was the top predictor of OBC (median = 30.2%; 95% CIs [20.5%, 69.1%]), followed by domestication status (median = 22.4%; 95% CIs [2.96%, 47.1%]) and geographical range (median = 11.1%; 95% CIs [5.09%, 21.9%]). The influence of taxonomic orders, traits, habitat and diet predictors varied per taxa of pathogen, and centrality measure studied as highlighted in figure 1 (electronic supplementary material, note S4 provides full details).

### (ii) Model performance metrics

Our models to explain centrality achieved the following performance: median $R^2 = 0.86$ (95% CIs [0.49, 0.97]); median adjusted $R^2 = 0.84$ [0.44, 0.97]; median normalized RMSE = 0.38 [0.06, 1.14] and median normalized MAE = 0.25 [0.03, 0.70]. Electronic supplementary material, note S4 lists full results for each centrality and pathogen taxa, and illustrates the performance of our ensembles compared with their based components.

## (c) Ensemble models to predict reservoirs of zoonoses and explain number of pathogens shared between humans and mammalian species

### (i) Predictors

Following averaging of variable importance with weights as per the previous section, centrality measures emerged as influential predictors across all our models (figures 2 and 3). Our novel II metric was the top single predictor across all models to determine if mammalian species harboured zoonoses or not (median = 30.1%; 95% CIs [8.67%, 63.10%]). ODC had the median influence of 13.6% [7.78%, 23.90%] and was second predictor in all models except DNA viruses (figure 2a). Centrality measures were also important in our models to explain the number of pathogens shared between humans and mammals. ODC (21.5%; [7.21%, 39.90%]) and OBC (20.9%; [10.50%, 28.10%]) were top predictors in eight of the nine models in this category, II significantly improved our helminth and bacterial models (figure 3a).

Entropy of pathogens research effort was the top predictor in our models to explain the number of zoonotic DNA viruses (median = 21.5%; [7.21%, 39.90%]). Overall, research

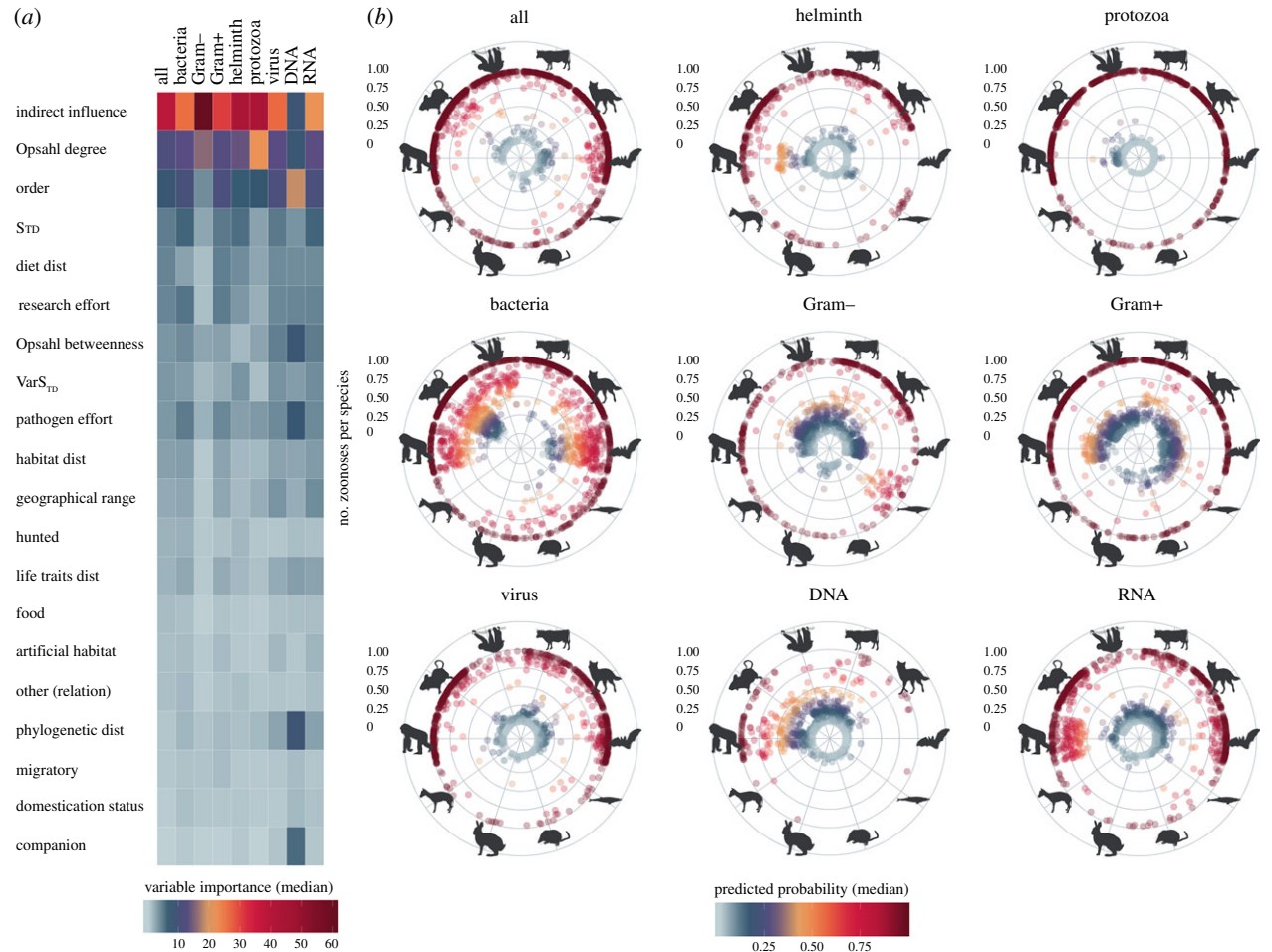

**Figure 2.** Results of ensemble models to predict reservoirs of zoonoses. (a) Median variable importance (relative influence) of predictors of reservoirs of zoonoses (based on the 100 runs of each model). For the purposes of this figure, the contribution of order predictors was summed. Details of contribution of all predictors of each model are presented in electronic supplementary material, Note S5. (b) Highlights predicted the median probability of host species in each order harbouring at least one zoonotic pathogen. Predictions were derived from all pathogens input set (i.e. mammalian hosts of any pathogen, n = 1560). Orders illustrated are as follows (clockwise): Artiodactyla, Carnivora, Chiroptera, Cetacea, Insectivora, Lagomorpha, Perissodactyla, Primates, Rodentia and other mammals (all remaining orders). (Online version in colour.)

effect (host and pathogen) influenced our models to explain number of zoonoses (pathogen = 9.05% [2.26%, 20.20%], host = 7.78% [5.20%, 10.40%]) nearly twice as much as our models to detect reservoirs of zoonoses (pathogen = 3.99% [1.75%, 8.00%], host = 4.31% [1.07%, 5.63%]).

The relative contribution of taxonomic orders, distances to human and domestication status to our models of reservoirs of zoonoses and number of pathogens shared with humans varied per pathogen taxa, electronic supplementary material, note S5 visualizes these differences.

### (ii) Predictions

We found that humans share pathogens with 1242 mammalian species; however, the number of species harbouring zoonoses differed per order and type of pathogen (figure 2b). More rodent species harboured zoonotic pathogens than any other order, except for DNA viruses, which were shared mainly with primates, and Gram+ bacteria, shared mainly with Artiodactyla and carnivores (81 species of each). Bats formed significant reservoirs of RNA viruses. Taking the total number of known extant species of species-rich orders into account [46], we found that 52.46% of carnivores shared pathogens with humans, followed by 35.50% of

known primates and 26.32% of known Artiodactyla. On the other hand, humans shared pathogens with only 15.44% and 13.91% of known bats and rodents, respectively.

Figure 3b illustrates the results of our models to predict and explain number of zoonoses shared with mammalian hosts. These results highlight differences in zoonoses sharing per host order and pathogen taxa, as well as variations in frequency of pathogen sharing with domesticated and wild species. For instance, humans tend to share helminths with carnivores (both domestic and wild). Zoonotic viruses, on the other hand, showed differences by genome type. DNA viruses are mostly shared with primates, whereas RNA viruses are more variedly shared with wild species particularly with bats, rodents and primates.

### (iii) Models performance metrics

We compiled a comprehensive set of performance metrics for all our models (electronic supplementary material, note S5). Our models to predict reservoirs of zoonoses had median AUC = 0.97 [0.94, 1.00], mean TSS = 0.71 [0.46, 1.00], mean KS statistics = 74.14 [62.31, 100.00] and mean F1-score = 0.95 [0.78, 0.99]. Our models to explain the number of pathogens shared with humans had the following performance metrics:

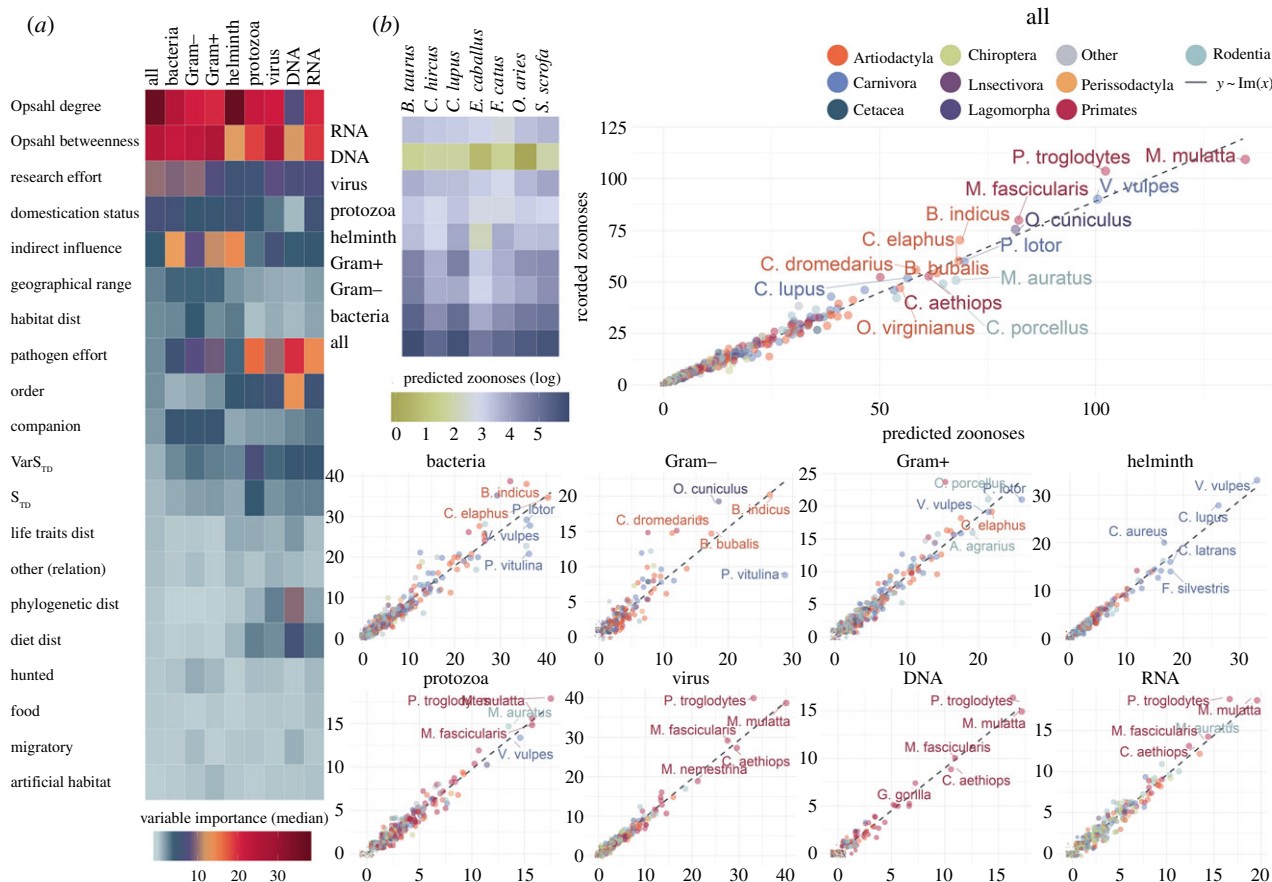

**Figure 3.** Results of our ensemble models to explain the number of zoonoses harboured by mammalian species. (*a*) Median variable importance (relative influence) of predictors included in the models over 100 runs. For the purposes of this figure, the contribution of order predictors is summed. Details of contribution of all predictors over the 100 runs of each model are presented in electronic supplementary material, note S6 and figure S12. (*b*) Predicted number of zoonoses in each mammalian host species. Heat-map illustrates logged predicted number of zoonoses in livestock, horses and domesticated dogs and cats. Points represent mammalian species, coloured by their order, size = log (predicted zoonoses + 1). *x*-axes are predicted number of zoonoses; *y*-axes are detected number of zoonoses. Labels are top species by number of predicted zoonoses (*n* = 10 in all pathogens panel, *n* = 5 in other panels). (Online version in colour.)

median $R^2 = 0.89$ [0.78, 0.94], median adjusted $R^2 = 0.89$ [0.76, 0.94], median normalized RMSE = 0.68 [0.39, 1.17] and median normalized MAE = 0.37 [0.23, 0.66]. Electronic supplementary material, note S5 lists the detailed performance metrics and compares the performance of our ensembles to their base learners.

## 4. Discussion

We presented a methodology integrating shared-pathogen networks and machine learning to answer three key questions: First, what makes a mammalian host species important in networks of shared pathogens? Second, which mammals are more likely to harbour zoonotic pathogens? Third, can the number zoonotic pathogens of a mammalian host be explained by its network centrality or host traits?

Our models to explain centrality (question 1) highlighted differences in traits associated with key species in each of our networks, as well as overarching general characteristics. Host species capable of sharing pathogens with distantly related species, and which harboured taxonomically varied pathogens, were more central due to being well connected with their neighbours (II and Opsahl degree). On the other hand, Opsahl betweenness was explained mainly by research effort, domestication status and geographical range. This indicated that there remain unexplored factors explaining bridge species harbouring pathogens shared between

otherwise loosely connected communities, for which research effort acted as a proxy.

Our models to predict and describe reservoirs of zoonoses (questions 2 and 3) confirmed that position in networks of shared pathogens is a key factor in determining the sharing of pathogens between mammals and humans; and extended this assumption to include all mammals rather than individual orders [9,12].

The work presented here builds on previous research [9,12,15,16,17,21] to advance our understanding of sharing of pathogens among mammalian sources of zoonoses in five key aspects: (i) our novel measure of centrality, II, while resembling closeness, is designed specifically for networks of shared pathogens. This measure provided key insight into the probability of mammalian hosts sharing pathogens with humans (question 2). It suggested that, regardless of pathogen type, mammal species that can spread more of their pathogens indirectly through their neighbours are more likely to harbour zoonoses than other species. (ii) Our ensembles provided a flexible, robust, unbiased mechanism to address our questions. (iii) By investigating sharing of pathogens across multiple taxa and sub-taxa (e.g. DNA/RNA viruses, Gram+/Gram− bacteria), we were able to ascertain important differences in the mechanisms of sharing of different types of pathogens, and the effect this has on zoonoses. (iv) In addition, by integrating three measures of centrality, rather than focusing on a

single measure, we were able to differentiate key species in networks based on the unique characteristics of each network. This was highlighted where host species with higher betweenness carried more zoonotic viruses and bacteria, whereas hosts with higher Opsahl degree carried more zoonotic helminths and protozoa. (v) Finally, we were also able to differentiate the roles that host order and their domestication status play in different networks.

We recognize several areas for future improvements of our models. First, our data were mined mainly from published research and deposited genetic sequences. While we made best effort to capture all interactions available within our resources [19] and to control for research effort as per previous studies [9,12,16], we realize there are inherent biases in our data sources. Recent research has revealed that between 20 and 40% of pathogen host ranges are currently unknown [47]. An area of future improvement will therefore focus on closing this particular knowledge gap, furthering recent attempts to predict missing interactions in networks such as ours. Second, edges in our networks, which represent sharing of pathogens, may be interpreted in a variety of ways including: spillover; direct contact; indirect sharing through vectors, intermediate hosts or environment; and coevolution of specific host–pathogen lineages. To further improve our models, we will aim to distinguish between these various events, focusing in particular on the role of transmission routes as means to include direction of pathogen sharing into our undirected networks. Third, we have focused our analyses on host and pathogen species interactions with minimal geographical layering. Geography plays a key role in facilitating host interactions and affects the probability of contact between humans and wildlife, which in turn increases chances of zoonotic transmission [5,48]. A future analysis will integrate geographical distribution of host species and will improve our ability to predict sources of zoonoses.

In summary, the work presented here highlighted differences in characteristics and centrality measures of networks of shared pathogens. It provided a methodology for selecting which centrality measure to include in analysis of similar networks. Our models revealed factors underlining centrality in these networks, and importantly how these factors vary across pathogen taxa. We established predictors of reservoirs of zoonoses and showcased centrality measures to be key in determining these reservoirs. Finally, we provided new insight into determinants of sharing of zoonoses between humans and mammalian hosts across major pathogen taxa.

Data accessibility. The data reported in this paper are accessible from the ENHanCEd Infectious Diseases (EID2) database at https://eid2.liverpool.ac.uk/. All datasets and R codes are made available via Figshare: https://doi.org/10.6084/m9.figshare.11537742, https://doi.org/10.6084/m9.figshare.11536470 and https://doi.org/10.6084/m9.figshare.11535279

Authors' contributions. M.B. and M.W. established the EID database. M.W. compiled data, designed, designed and ran the analyses. K.J.S. provided expertise into centrality analyses. All authors contributed to interpretation of the results and writing of manuscripts and electronic supplementary material.

Competing interests. We declare we have no competing interests.

Funding. M.W. acknowledges support from RCUK/UKRI Innovation Fellowship (grant no. MR/R024898/1). Establishment of the EID2 database was funded by a UK Research Council Grant (grant no. NE/G002827/1) to M.B., as part of an ERANET Environmental Health award to M.B.; subsequently, it has been further developed and maintained by BBSRC Tools and Resources Development Fund awards (BB/K003798/1; BB/N02320X/1) to M.B., and the National Institute for Health Research Health Protection Research Unit (NIHR HPRU) in Emerging and Zoonotic Infections at the University of Liverpool in partnership with Public Health England and Liverpool School of Tropical Medicine.

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
