## [Reviewer comments · Proceedings of the Royal Society B: Biological Sciences]

Review History

RSPB-2019-2062.R0 (Original submission)

Review form: Reviewer 1

Recommendation

Major revision is needed (please make suggestions in comments)

Scientific importance: Is the manuscript an original and important contribution to its field?

Acceptable

General interest: Is the paper of sufficient general interest?

Acceptable

Quality of the paper: Is the overall quality of the paper suitable?

Marginal

Is the length of the paper justified?

Yes

Should the paper be seen by a specialist statistical reviewer?

No

Do you have any concerns about statistical analyses in this paper? If so, please specify them explicitly in your report.

Yes

It is a condition of publication that authors make their supporting data, code and materials available - either as supplementary material or hosted in an external repository. Please rate, if applicable, the supporting data on the following criteria.

Is it accessible?

Yes

Is it clear?

Yes

Is it adequate?

Yes

Do you have any ethical concerns with this paper?

No

Comments to the Author

The manuscript by Wardeh et al. presents a data-driven approach to explain the structure of mammalian shared pathogen networks, which includes quantitative comparisons of various machine learning algorithms, a comparison of network statistics that characterize structure, and the introduction of a new metric for node influence "Indirect Influence".

Overall, the paper addresses an important set of questions about host-pathogen networks, but I have concerns about the information lost in focusing on the unipartite (host-host) projection of the underlying bipartite (host-symbiont) network, and about circularity in the zoonotic reservoir prediction approach.

Major issues

1. The paper focuses on the host-host pathogen sharing network, which represents a unipartite projection of the underlying host-parasite bipartite network. This unipartite projection results in an information loss - namely, in computing the unipartite projection, information about which particular symbionts infect each host is discarded. The splitting of the analysis by symbiont type (Helminth, Protozoa, Bacteria, etc.) provides only a partial solution: within any one of these types, the unipartite host-host network contains no information about which helminths, protozoa, bacteria etc. are shared.

While an analysis of the bipartite network would be ideal in my mind, at least acknowledging that the host-host network is a summary of the underlying bipartite network is critical. What are the implications of lumping together all protozoa, for example? And, if subsequent work wanted to extend this paper to a bipartite case, what would that look like?

2. The prediction of zoonosis reservoirs suggests that the new network metric "Indirect Influence" was an important covariate, followed by Opsahl degree centrality. But, this analysis seems to be circular: to predict whether or not a host species is a reservoir, you would first need to compute Indirect Influence and Opsahl degree, which require that the network is observed. If the network is observed, then you would already know whether a species is a zoonotic reservoir. The input to the model for whether a node in the host-host network is a zoonotic reservoir consists of node-level summaries of the host-host network, which is circular.

To see why this is circular, consider how one would make a prediction for a new host species, for which no host-symbiont data were available (e.g., a new node in the host-host network). With no

data about that node, it would be impossible to compute II and Opsahl degree centrality, which are inputs to the predictive model.

Minor issues

Line 357: The term "superspreaders" usually refers to individuals in a population that disproportionately transmit disease (e.g., as described in Lloyd-Smith et al. 2005, *Nature*), but here it seems to refer to host species that are well-connected in the host-host pathogen sharing network. I'd recommend against using this term here, unless you can cite other papers that are using it in this context.

For the model of the number of pathogens shared between humans and mammalian species, have you considered a response distribution that has integer support? e.g., a Poisson or negative binomial distribution? If the counts are very large on average, a normal approximation is likely to work just as well, but if the counts tend to be small (or often 0), a count model might work better.

Line 357-360: The discussion of the Opsahl betweenness results seems to minimize the finding that reporting effort was the more important predictor overall. I give the authors credit for acknowledging that the networks are incompletely observed later in the discussion, but it does seem odd to ignore this effect in these lines. In particular, readers are likely to be curious why well-studied species might "act as bridge species harbouring pathogens shared between otherwise loosely connected communities".

The writing could be improved in places:

Line 28: The first sentence of the abstract seems to be missing a comma after "zoonoses".

Line 35: Should "life traits" be "life-history traits"?

Line 36-37: The word "explaining" might be more precise than "drivers underpinning".

Line 40: It would be more correct to write "to be a key factor" than "to be key factor".

Line 59: The last sentence of the first paragraph seems to have some extra words: "...so that we can anticipate possible outbreaks in advance anticipated (8, 9)."

Line 108-109: I would recommend removing the word "approach" in "we utilise principle component analysis approach (PCA)...".

Line 122: The word "flavours" is somewhat colloquial. Is there a more precise word that would be better? e.g., "types"?

Line 136-138, equations 1 - 4: The mathematical writing could be improved. First, using punctuation with equations can improve readability. For example, you might add commas or periods as appropriate after numbered equations depending on whether the sentence continues. Second, what are the $|$ symbols representing in equations 2-4? I would expect these to be absolute value or determinant symbols, but it's not clear from the text what they represent. You might see the MEE paper "Some guidance on using mathematical notation in ecology" by Andrew M. Edwards and Marie Auger-Méthé 2018 for some specific advice: <https://besjournals.onlinelibrary.wiley.com/doi/10.1111/2041-210X.13105>

Review form: Reviewer 2

Recommendation

Major revision is needed (please make suggestions in comments)

Scientific importance: Is the manuscript an original and important contribution to its field?

Excellent

General interest: Is the paper of sufficient general interest?

Good

Quality of the paper: Is the overall quality of the paper suitable?

Marginal

Is the length of the paper justified?

Yes

Should the paper be seen by a specialist statistical reviewer?

Yes

Do you have any concerns about statistical analyses in this paper? If so, please specify them explicitly in your report.

Yes

It is a condition of publication that authors make their supporting data, code and materials available - either as supplementary material or hosted in an external repository. Please rate, if applicable, the supporting data on the following criteria.

Is it accessible?

N/A

Is it clear?

N/A

Is it adequate?

N/A

Do you have any ethical concerns with this paper?

No

Comments to the Author

The manuscript by Wardeh et al addresses an interesting and hugely important topic, the arrival of new zoonoses in the human population. Specifically the authors tackle the task of identifying important species in the network where nodes are species. They do this by considering the networks of shared pathogens between species. After a preliminary analysis to identify suitable network metric the authors then use machine learning to identify traits that affect the species propensity to harbor potential zoonoses and also identify particular species that are likely to be transferred to humans.

This topic is of great importance and will be of interest to a very broad readership ranging from ecology and epidemiology to microbiology and medicine. The subject matter is therefore well worthy of publication Proceedings B. The authors tackle the overarching question with a novel and interesting approach. However, due to some particular choices that have been made and also issues with the write up the results are much less convincing than they could be.

I have four major issues with the present form of the manuscript in its present form:

1) First, this paper was a tough read, both generally speaking and particular by the standards of Proceedings B. In terms of grammar and orthography the writing is fine. The overall organization into sections, where methods are separated often complicates the presentation of theory papers, and this one is no exception to this rule. However, my main problem with the text was that it is ripe with numerical factoids, but lacks the context to put them into perspective. In other words it's hard to see the forest for the trees. This style is very common in experimental biology papers and also in some conceptually simple data analysis, where the basics of the approach are clear. However, in a complex study like the present one it conceals the major lines of thinking. I found myself endlessly rereading paragraphs in search for certain bits of information. Also it was very hard to connect the headline messages to the actual nuts and bolts. I come back to this point several times below.

2) Perhaps because of the above, my central issue with this manuscript is that I am not convinced the results are informative. The manuscript tackles deep and difficult questions for which little ground truth is available. Any faith in the validity of the result must thus stem from trust in the procedures by which the results are obtained. For instance reading the main text I had no clear understanding of how the network metrics were selected, this was only a little cleared up by considering the supporting files. The authors use a combination of PCA and k-means for this task. In my assessment these are blunt outdated tools and it has been well demonstrated that they can fail spectacularly. Today there are much better methods for these tasks: diffusion maps, umap that deal more gracefully with nonlinearity and high-dimensional data sets.

The machine learning methods are closer than state of the art, however the methods listed are classifiers and it is not clear to me in how far the problem that is addressed here is a classification problem? This is probably related to my inability to understand the red line in the reasoning, described above.

3) The manuscript tries to convince the reader using statistical arguments. Admittedly, my knowledge of statistics may be insufficient to fully appreciate the abundance of performance metrics presented, but many readers will feel the same. What I am looking for is a compelling argument why I should trust the results. The information I get is that the model "explains" much of the variation in the data, but given the size of the search space this alone is not a compelling argument. My understanding is that the statistics fall short of giving us actual error bounds on the results, or is this not the case?

4) Perhaps because of the statistical / empirical approach the manuscript is almost free of "theory" and hence theoretical points that need to be discussed remain unaddressed.

My central concern is the following: The main motivating idea that runs through the introduction is to identify reservoirs of future zoonosis for humans. It is not clear to me that this information is in principle discoverable by analysing the network of pathogens that have been shared in the past. The analysis of traits and habitat factors etc is promising, but in my reading the final step that needs to be made, essentially niche modelling, does not get across clearly.

I was actually waiting for the manuscript to make this final arc but then it took the wrong turn. The conclusions seem to highlight domesticated species as the ones we share most pathogens with. This is clearly true, but seems to forget the central question. Equally clearly, domesticated mammals are unlikely sources of undiscovered zoonoses, as we have already been in contact with these species for a long time.

Specific comments:

- When I hear machine learning I am nowadays expecting that the words deep neural networks and TensorFlow follow. But hear this wasn't done, is there a reason.

- Why is it actually necessary to boil down the metrics before the rest of the analysis is done? Surely the machine learning approaches can deal with multiple metrics?
 - Is there actually a message from the network plots in figure 1?
 - The "Opsahl" centralities have a free parameter alpha. How was this chosen? And, why?
 - Why is there a need for the new indirect influence? Aren't there enough network metrics already? It seems included as an afterthought after the selection of the other metrics, is this the case? Was this necessary to produce meaningful results? Why this particular form? Is there a mechanistic reasoning rather than looking being reminiscent of entropy?
 It seems very odd to find this specific insertion in a paper that otherwise goes on a broad trawl of methods.

Let me finish by admitting that I have struggled with this manuscript quite a bit. Please accept my apologies for being very slow with review and likely misinterpreting some important points.

Decision letter (RSPB-2019-2062.R0)

14-Oct-2019

Dear Dr Wardeh:

I am writing to inform you that your manuscript RSPB-2019-2062 entitled "Integration of shared-pathogen networks and machine learning reveal key aspects of zoonoses and predict mammalian reservoirs" has, in its current form, been rejected for publication in Proceedings B.

This action has been taken on the advice of referees, who have recommended that substantial revisions are necessary. With this in mind we would be happy to consider a resubmission, provided the comments of the referees are fully addressed. However please note that this is not a provisional acceptance.

Sincerely,
 Dr Sasha Dall
 mailto: proceedingsb@royalsociety.org

Associate Editor

Comments to Author:

Both reviewers consider this work as important but they have major concerns in several areas; robustness and sensitive of the tools used in the analysis of networks in predicting zoonosis. Strikingly, both reviewers identify critical species interactions not addressed in the study. After reading the manuscript, I fully agree with the concerns of both author.

Reviewer(s)' Comments to Author:

Referee: 1

Comments to the Author(s)

The manuscript by Wardeh et al. presents a data-driven approach to explain the structure of mammalian shared pathogen networks, which includes quantitative comparisons of various machine learning algorithms, a comparison of network statistics that characterize structure, and the introduction of a new metric for node influence "Indirect Influence".

Overall, the paper addresses an important set of questions about host-pathogen networks, but I have concerns about the information lost in focusing on the unipartite (host-host) projection of the underlying bipartite (host-symbiont) network, and about circularity in the zoonotic reservoir prediction approach.

Major issues

1. The paper focuses on the host-host pathogen sharing network, which represents a unipartite projection of the underlying host-parasite bipartite network. This unipartite projection results in an information loss - namely, in computing the unipartite projection, information about which particular symbionts infect each host is discarded. The splitting of the analysis by symbiont type (Helminth, Protozoa, Bacteria, etc.) provides only a partial solution: within any one of these types, the unipartite host-host network contains no information about which helminths, protozoa, bacteria etc. are shared.

While an analysis of the bipartite network would be ideal in my mind, at least acknowledging that the host-host network is a summary of the underlying bipartite network is critical. What are the implications of lumping together all protozoa, for example? And, if subsequent work wanted to extend this paper to a bipartite case, what would that look like?

2. The prediction of zoonosis reservoirs suggests that the new network metric "Indirect Influence" was an important covariate, followed by Opsahl degree centrality. But, this analysis seems to be circular: to predict whether or not a host species is a reservoir, you would first need to compute Indirect Influence and Opsahl degree, which require that the network is observed. If the network is observed, then you would already know whether a species is a zoonotic reservoir. The input to the model for whether a node in the host-host network is a zoonotic reservoir consists of node-level summaries of the host-host network, which is circular.

To see why this is circular, consider how one would make a prediction for a new host species, for which no host-symbiont data were available (e.g., a new node in the host-host network). With no data about that node, it would be impossible to compute II and Opsahl degree centrality, which are inputs to the predictive model.

Minor issues

Line 357: The term "superspreaders" usually refers to individuals in a population that disproportionately transmit disease (e.g., as described in Lloyd-Smith et al. 2005, Nature), but here it seems to refer to host species that are well-connected in the host-host pathogen sharing

network. I'd recommend against using this term here, unless you can cite other papers that are using it in this context.

For the model of the number of pathogens shared between humans and mammalian species, have you considered a response distribution that has integer support? e.g., a Poisson or negative binomial distribution? If the counts are very large on average, a normal approximation is likely to work just as well, but if the counts tend to be small (or often 0), a count model might work better.

Line 357-360: The discussion of the Opsahl betweenness results seems to minimize the finding that reporting effort was the more important predictor overall. I give the authors credit for acknowledging that the networks are incompletely observed later in the discussion, but it does seem odd to ignore this effect in these lines. In particular, readers are likely to be curious why well-studied species might "act as bridge species harbouring pathogens shared between otherwise loosely connected communities".

The writing could be improved in places:

Line 28: The first sentence of the abstract seems to be missing a comma after "zoonoses".

Line 35: Should "life traits" be "life-history traits"?

Line 36-37: The word "explaining" might be more precise than "drivers underpinning".

Line 40: It would be more correct to write "to be a key factor" than "to be key factor".

Line 59: The last sentence of the first paragraph seems to have some extra words: "...so that we can anticipate possible outbreaks in advance anticipated (8, 9)."

Line 108-109: I would recommend removing the word "approach" in "we utilise principle component analysis approach (PCA)...".

Line 122: The word "flavours" is somewhat colloquial. Is there a more precise word that would be better? e.g., "types"?

Line 136-138, equations 1 - 4: The mathematical writing could be improved. First, using punctuation with equations can improve readability. For example, you might add commas or periods as appropriate after numbered equations depending on whether the sentence continues. Second, what are the $|$ $|$ symbols representing in equations 2-4? I would expect these to be absolute value or determinant symbols, but it's not clear from the text what they represent. You might see the MEE paper "Some guidance on using mathematical notation in ecology" by Andrew M. Edwards and Marie Auger-Méthé 2018 for some specific advice: <https://besjournals.onlinelibrary.wiley.com/doi/10.1111/2041-210X.13105>

Referee: 2

Comments to the Author(s)

The manuscript by Wardeh et al addresses an interesting and hugely important topic, the arrival of new zoonoses in the human population. Specifically the authors tackle the task of identifying important species in the network where nodes are species. They do this by considering the networks of shared pathogens between species. After a preliminary analysis to identify suitable network metric the authors then use machine learning to identify traits that affect the species propensity to harbor potential zoonoses and also identify particular species that are likely to be transferred to humans.

This topic is of great importance and will be to interest to a very broad readership ranging from

ecology and epidemiology to microbiology and medicine. The subject matter is therefore well worthy of publication Proceedings B. The authors tackle it the overarching question with a novel and interesting approach. However, due to some particular choices that have been made and also issues with the write up the results are much less convincing that they could be.

I have four major issues with the present form of the manuscript in its present form:

1) First, this paper was a tough read, both generally speaking and particular by the standards of Proceedings B. In terms of grammar and orthography the writing is fine. The overall organization into sections, where methods are separated often complicates the presentation of theory papers, and this one is no exception to this rule. However, my main problem with the text was that it is ripe with numerical factoids, but lacks the context to put them into perspective. In other words it's hard to see the forest for the trees. This style is very common in experimental biology papers and also in some conceptually simple data analysis, where the basics of the approach are clear. However, in a complex study like the present one it conceals the major lines of thinking. I found myself endlessly rereading paragraphs in search for certain bits of information. Also it was very hard to connect the headline messages to the actual nuts and bolts. I come back to this point several times below.

2) Perhaps because of the above, my central issue with this manuscript is that I am not convinced the results are informative. The manuscript tackles deep and difficult questions for which little ground truth is available. Any faith in the validity of the result must thus stem from trust in the procedures by which the results are obtained. For instance reading the main text I had no clear understanding of how the network metrics were selected, this was only a little cleared up by considering the supporting files. The authors use a combination of PCA and k-means for this task. In my assessment these are blunt outdated tools and it has been well demonstrated that they can fail spectacularly. Today there are much better methods for these tasks: diffusion maps, umap that deal more gracefully with nonlinearity and high-dimensional data sets.

The machine learning methods are closer than state of the art, however the methods listed are classifiers and it is not clear to me in how far the problem that is addressed here is a classification problem? This is probably related to my inability to understand the red line in the reasoning, described above.

3) The manuscript tries to convince the reader using statistical arguments. Admittedly, my knowledge of statistics may be insufficient to fully appreciate the abundance performance metrics presented, but many readers will feel the same. What I am looking for is a compelling argument why I should trust the results. The information I get is that the model "explains" much of the variation in the data, but given the size of the search space this alone is not a compelling argument. My understanding is that the statistics fall short of giving us actual error bounds on the results, or is this not the case?

4) Perhaps because of the statistical / empirical approach the manuscript is almost free of "theory" and hence theoretical points that need to be discussed remain unaddressed.

My central concern is the following: The main motivating idea that runs through the introduction is to identify reservoirs of future zoonosis for humans. It is not clear to me that this information is in principle discoverable by analysing the network of pathogens that have been shared in the past. The analysis of traits and habitat factors etc is promising, but in my reading the final step that needs to be made, essentially niche modelling, does not get across clearly.

I was actually waiting for the manuscript to make this final arc but then it took the wrong turn. The conclusions seem to highlight domesticated species as the ones we share most pathogens with. This is clearly true, but seems to forget the central question. Equally clearly, domesticated mammals are unlikely sources of undiscovered zoonoses, as we have already been in contact with these species for a long time.

Specific comments:

- When I hear machine learning I am nowadays expecting that the words deep neural networks and TensorFlow follow. But here this wasn't done, is there a reason.
 - Why is it actually necessary to boil down the metrics before the rest of the analysis is done? Surely the machine learning approaches can deal with multiple metrics?
 - Is there actually a message from the network plots in figure 1?
 - The "Opsahl" centralities have a free parameter alpha. How was this chosen? And, why?
 - Why is there a need for the new indirect influence? Aren't there enough network metrics already? It seems included as an afterthought after the selection of the other metrics, is this the case? Was this necessary to produce meaningful results? Why this particular form? Is there a mechanistic reasoning rather than looking being reminiscent of entropy?
- It seems very odd to find this specific insertion in a paper that otherwise goes on a broad trawl of methods.

Let me finish by admitting that I have struggled with this manuscript quite a bit. Please accept my apologies for being very slow with review and likely misinterpreting some important points.

Author's Response to Decision Letter for (RSPB-2019-2062.R0)

See Appendix A.

RSPB-2019-2882.R0

Review form: Reviewer 1

Recommendation

Accept as is

Scientific importance: Is the manuscript an original and important contribution to its field?

Acceptable

General interest: Is the paper of sufficient general interest?

Acceptable

Quality of the paper: Is the overall quality of the paper suitable?

Acceptable

Is the length of the paper justified?

Yes

Should the paper be seen by a specialist statistical reviewer?

No

Do you have any concerns about statistical analyses in this paper? If so, please specify them explicitly in your report.

No

It is a condition of publication that authors make their supporting data, code and materials available - either as supplementary material or hosted in an external repository. Please rate, if applicable, the supporting data on the following criteria.

Is it accessible?

Yes

Is it clear?

Yes

Is it adequate?

Yes

Do you have any ethical concerns with this paper?

No

Comments to the Author

The manuscript by Wardeh et al. presents a data-driven approach to explain the structure of mammalian shared pathogen networks, which includes quantitative comparisons of various machine learning algorithms, a comparison of network statistics that characterize structure, and new network metrics. The authors have done a great job addressing my comments from the original submission. The revised manuscript is more clear, and I have no additional major comments.

Minor suggestions

In figure 4, a dashed line along the diagonal for the scatter plots might be better than the smooth fits that are currently presented, which seem to be sensitive to outliers (e.g., the Gram - panel). This is a very minor (and mostly aesthetic) comment, however.

Review form: Reviewer 2

Recommendation

Accept as is

Scientific importance: Is the manuscript an original and important contribution to its field?

Good

General interest: Is the paper of sufficient general interest?

Excellent

Quality of the paper: Is the overall quality of the paper suitable?

Good

Is the length of the paper justified?

Yes

Should the paper be seen by a specialist statistical reviewer?

No

Do you have any concerns about statistical analyses in this paper? If so, please specify them explicitly in your report.

No

It is a condition of publication that authors make their supporting data, code and materials available - either as supplementary material or hosted in an external repository. Please rate, if applicable, the supporting data on the following criteria.

Is it accessible?

N/A

Is it clear?

N/A

Is it adequate?

N/A

Do you have any ethical concerns with this paper?

No

Comments to the Author

The revision has addressed my concerns to some extent. I am still not 100% confident in the results, but this is unlikely to improve with further revisions. The manuscript takes an interesting approach to an interesting topic and is not misleading. I am therefore in favour of publication.

Decision letter (RSPB-2019-2882.R0)

03-Jan-2020

Dear Dr Wardeh

I am pleased to inform you that your manuscript RSPB-2019-2882 entitled "Integration of shared-pathogen networks and machine learning reveal key aspects of zoonoses and predict mammalian reservoirs" has been accepted for publication in Proceedings B.

The referees have recommended publication, but also suggest some minor revisions to your manuscript. Therefore, I invite you to respond to the comments and revise your manuscript. Because the schedule for publication is very tight, it is a condition of publication that you submit the revised version of your manuscript within 7 days. If you do not think you will be able to meet this date please let us know.

When submitting your revised manuscript, you will be able to respond to the comments made by the referee(s) and upload a file "Response to Referees". You can use this to document any changes you make to the original manuscript. We require a copy of the manuscript with revisions made

since the previous version marked as 'tracked changes' to be included in the 'response to referees' document.

Sincerely,
Dr Sasha Dall
mailto: proceedingsb@royalsociety.org

Reviewer(s)' Comments to Author:

Referee: 2

Comments to the Author(s).

The revision has addressed my concerns to some extent. I am still not 100% confident in the results, but this is unlikely to improve with further revisions. The manuscript takes an interesting approach to an interesting topic and is not misleading. I am therefore in favour of publication.

Referee: 1

Comments to the Author(s).

The manuscript by Wardeh et al. presents a data-driven approach to explain the structure of mammalian shared pathogen networks, which includes quantitative comparisons of various machine learning algorithms, a comparison of network statistics that characterize structure, and new network metrics. The authors have done a great job addressing my comments from the original submission. The revised manuscript is more clear, and I have no additional major comments.

Minor suggestions

In figure 4, a dashed line along the diagonal for the scatter plots might be better than the smooth fits that are currently presented, which seem to be sensitive to outliers (e.g., the Gram - panel). This is a very minor (and mostly aesthetic) comment, however.

Author's Response to Decision Letter for (RSPB-2019-2882.R0)

See Appendix B.

Decision letter (RSPB-2019-2882.R1)

09-Jan-2020

Dear Dr Wardeh

I am pleased to inform you that your manuscript entitled "Integration of shared-pathogen networks and machine learning reveal key aspects of zoonoses and predict mammalian reservoirs" has been accepted for publication in Proceedings B.

Open Access

Paper charges

Sincerely,

Proceedings B

Appendix A

Response to referee 1

The manuscript by Wardeh et al. presents a data-driven approach to explain the structure of mammalian shared pathogen networks, which includes quantitative comparisons of various machine learning algorithms, a comparison of network statistics that characterize structure, and the introduction of a new metric for node influence "Indirect Influence".

Overall, the paper addresses an important set of questions about host-pathogen networks, but I have concerns about the information lost in focusing on the unipartite (host-host) projection of the underlying bipartite (host-symbiont) network, and about circularity in the zoonotic reservoir prediction approach.

We thank referee 1 for their efforts reading our paper and their insightful comments. We have tried to address their concerns and answer their comments in as much detail as possible.

Major issues

1. The paper focuses on the host-host pathogen sharing network, which represents a unipartite projection of the underlying host-parasite bipartite network. This unipartite projection results in an information loss - namely, in computing the unipartite projection, information about which particular symbionts infect each host is discarded. The splitting of the analysis by symbiont type (Helminth, Protozoa, Bacteria, etc.) provides only a partial solution: within any one of these types, the unipartite host-host network contains no information about which helminths, protozoa, bacteria etc. are shared. While an analysis of the bipartite network would be ideal in my mind, at least acknowledging that the host-host network is a summary of the underlying bipartite network is critical. What are the implications of lumping together all protozoa, for example? And, if subsequent work wanted to extend this paper to a bipartite case, what would that look like?

Our response: The referee raises a number of important points. We agree that loss of some information is inevitable when projecting from bipartite to unipartite networks. In fact, a major motivation behind our Indirect Influence metric was to address this issue by retaining vital

information about the pathogens shared between host species. We have amended the text of the main manuscript in two locations to address this issue, and highlight the effect of implications of lumping together all pathogens of given taxa (e.g. protozoa).

With regards to subsequent work extending the paper to bipartite networks we believe it will depend on the intention behind this extension: whether to investigate sharing of pathogens amongst hosts in relation to zoonoses or prediction of host-pathogen interactions. For the purposes of the first task, which was the focus of this paper, we believe an adaptation of common centrality measures and network statistics to bipartite networks will be required. Values of our indirect influence measure however, will remain the same, as they have, in effect, been calculated in the bipartite network.

Changes: We have amended our methods section in two locations as highlighted below.

Subsection: Host-pathogen species interactions and network formulation (lines 78-94) – new text underlined:

We extracted interactions between non-human mammals and their pathogens from the Enhanced Infectious Diseases Database (EID2) (23). These interactions formed a bipartite network with nodes represented hosts and pathogens, and links indicated whether pathogens have been found in hosts. We further checked these interactions to ascertain whether the putative pathogens caused a disease or an opportunistic infection in at least one species of mammal (including humans). This resulted in 16,548 species-level host-pathogen interactions between 3,986 pathogen species (bacteria=885, fungi=251, helminth=1000, protozoa=404, and virus=1446) and 1,560 non-human mammalian species.

We projected this bipartite network into a unipartite network where nodes represented host species and edges quantified shared pathogens. The motivation behind this projection is twofold: 1) it enabled us to investigate dynamics of pathogen sharing via ecological network analysis tools (10, 24); and 2) it facilitated identification of important non-human mammalian species via centrality measures. In addition to the network encompassing all pathogen taxa (including fungi), we generated eight additional networks: bacteria (including Gram variable), Gram+ bacteria, Gram- bacteria, helminths, protozoa, viruses (including retro-transcribing), DNA viruses and RNA viruses. Prions were not included in this study (Supplementary Note 1 lists further information).

Subsection: Novel metric of node Influence – Indirect Influence (II) (lines 132- 155):

Unipartite projection of host-pathogen bipartite networks results in inevitable loss of information (39).

Let us assume, for instance, that we have three mammalian species: A, B and C, and that A and B share 10 protozoan agents, whereas B and C share 5 protozoan agents. Here we have two extreme scenarios: 1) The protozoans are all different; A and C do not share any protozoan pathogens, which in turns means that there is no flow of protozoan pathogens from A to C despite a path appearing to exist between the two (via B); 2) The 5 protozoans shared by B and C are among the 10 shared by A and B and, therefore, half of A's protozoans are shared with C, and all of C's are shared with A. Both scenarios have implications on the traditional centrality metrics analysed in the previous subsection.

To address these issues we developed a novel entropy-based metric, which we term Indirect Influence (II). II captures the influence each host species exercises within the unipartite network by measuring the number and frequency of pathogens this host spreads indirectly through its neighbouring species if it were to interact with each of them in isolation to the remainder of the network. Entropy enables us to capture uncertainties in the destination of pathogens as a function of the original host (40) at three levels - i) a species which share a few pathogens with many neighbours: entropy captures uncertainty in which neighbour it could influence indirectly; ii) a species which shares many pathogens with few neighbours: entropy allows us to capture uncertainty in pathogens shared; iii) a species which shares many pathogens with many neighbours will have high centrality due to uncertainties in both pathogens shared and neighbours influenced. In addition, using entropy allowed us to avoid assessing the many paths connecting all node pairs and, instead, focus on the potential of the node (i.e. the host species) to diversify pathogen propagation (40,41).

New references:

39. Zhou T, Ren J, Medo M, Zhang YC. Bipartite network projection and personal recommendation. *Phys Rev E - Stat Nonlinear, Soft Matter Phys.* 2007; 76(4).

40. Nikolaev AG, Razib R, Kucheriya A. On efficient use of entropy centrality for social network analysis and community detection. *Soc Networks.*2015; 40:154–62.

2. The prediction of zoonosis reservoirs suggests that the new network metric "Indirect Influence" was an important covariate, followed by Opsahl degree centrality. But, this analysis seems to be circular: to predict whether or not a host species is a reservoir, you would first need to compute Indirect Influence and Opsahl degree, which require that the network is observed. If the network is observed, then you would already know whether a species is a zoonotic reservoir. The input to the model for whether a node in the host-host network is a zoonotic reservoir consists of node-level summaries of the host-host network, which is circular.

To see why this is circular, consider how one would make a prediction for a new host species, for which no host-symbiont data were available (e.g., a new node in the host-host network). With no data about that node, it would be impossible to compute II and Opsahl degree centrality, which are inputs to the predictive model.

Our response: we thank the referee for raising these crucial points. Below we address them in order.

1) **Circularity:** we argue that observing the network of shared pathogens amongst non-human mammals does not automatically mean we know (for certain) that the hosts are carriers of zoonoses or not. This is for two reasons: i) the data are not perfect; while we have information on many reservoirs of zoonoses, we likely don't know all of them. By observing networks of shared pathogens amongst non-human mammals (i.e. centrality measures), and collating various distances from humans, we were able to detect potential reservoirs of zoonoses that were not apparent in the underlying dataset. ii) The networks we observed did not include humans, and the centralities calculated therefore summarised aspects of sharing of these pathogens among non-human mammals. In other words, as the networks we observed consisted only of non-human mammalian hosts, the importance of hosts in these networks are not necessarily a product of them being reservoirs of zoonoses. For instance, livestock have prominent position in the network due to sharing of livestock-only pathogens with other domestic and wild species, rather than being reservoirs of zoonoses. Our models, therefore, are best seen as explanatory tools of traits and features associated with zoonoses.

2) **Mammals without pathogens:** We agree that our models are not designed to handle mammalian species with no available information of their position in the network (i.e. no information on pathogens in species). We wanted to test the relationship between how non-human mammals share pathogens with each other and their status as reservoirs of zoonoses. However, where no data are available on pathogens within a mammalian species, this species can still enter the models with centralities all equal to 0 (similar to known hosts which do not share any of their pathogens with any other hosts).

Changes: To highlight the issues raised above, we updated Figure 3 with predictions performed with input set including all mammalian hosts. This figure now showcase risk (as probability between 0 and 1) of each mammalian species found to be a host of at least one species of any type of pathogen (i.e. the dataset included in all pathogens models). We attach the figure and its legend below.

Figure 3 – Results of our ensemble models to predict reservoirs of zoonoses. Panel A: median variable importance (relative influence) of predictors of reservoirs of zoonoses (based on the 100 runs of each model). For the purposes of this figure the contribution of order predictors were summed. Details of contribution of all predictors of each model are presented in Supplementary Note 5. **Panel**

B: highlights predicted median probability of host species in each order harbouring at least one zoonotic pathogen. Predictions were derived from all pathogens input set (i.e. mammalian hosts of any pathogen, N=1,560). Orders illustrated are as follows (clockwise): Artiodactyla, Carnivora, Chiroptera, Cetacea, Insectivora, Lagomorpha, Perissodactyla, Primates, Rodentia, and other mammals (all remaining orders).

Minor issues:

Line 357: The term "superspreaders" usually refers to individuals in a population that disproportionately transmit disease (e.g., as described in Lloyd-Smith et al. 2005, Nature), but here it seems to refer to host species that are well-connected in the host-host pathogen sharing network. I'd recommend against using this term here, unless you can cite other papers that are using it in this context.

We agree with the referee. We changed the term "super-spreaders" to "well-connected".

Changes - Discussion (lines 344- 347):

Host species capable of sharing pathogens with distantly related species, and which harboured taxonomically varied pathogens, were more central due to being well-connected with their neighbours (Indirect Influence and Opsahl degree)

For the model of the number of pathogens shared between humans and mammalian species, have you considered a response distribution that has integer support? e.g., a Poisson or negative binomial distribution? If the counts are very large on average, a normal approximation is likely to work just as well, but if the counts tend to be small (or often 0), a count model might work better.

The models used are robust to count data (glmnet, gbm, random forests, rpart, and svm with radial basis) as well as the overall ensemble (variation of glm). As we have wanted to use the same set of models for all our analyses we maintained the same setup for the three experiments performed. A future improvement might be to select best performing models for each iteration/problem from a larger set of models.

Line 357-360: The discussion of the Opsahl betweenness results seems to minimize the finding that reporting effort was the more important predictor overall. I give the authors credit for acknowledging that the networks are incompletely observed later in the discussion, but it does seem odd to ignore this effect in these lines. In particular, readers are likely to be curious why well-studied species might "act as bridge species harbouring pathogens shared between otherwise loosely connected communities".

We agree with the referee, we have amended our discussion so the role of research effort is addressed.

The changes are listed below.

Changes: Discussion (lines 345-349):

On the other hand, Opsahl betweenness was explained mainly by research effort, domestication status and geographical range. This indicated that there remain unexplored factors explaining bridge species harbouring pathogens shared between otherwise loosely connected communities, for which research effort acted as a proxy.

The writing could be improved in places:

Line 28: The first sentence of the abstract seems to be missing a comma after "zoonoses".

Line 35: Should "life traits" be "life-history traits"?

Line 36-37: The word "explaining" might be more precise than "drivers underpinning".

Line 40: It would be more correct to write "to be a key factor" than "to be key factor".

Line 59: The last sentence of the first paragraph seems to have some extra words: "...so that we can anticipate possible outbreaks in advance anticipated (8, 9)."

Line 108-109: I would recommend removing the word "approach" in "we utilise principle component analysis approach (PCA)...".

Line 122: The word "flavours" is somewhat colloquial. Is there a more precise word that would be better? e.g., "types"?

Line 136-138, equations 1 - 4: The mathematical writing could be improved. First, using punctuation with equations can improve readability. For example, you might add commas or periods as appropriate after numbered equations depending on whether the sentence continues. Second, what are

the $||$ symbols representing in equations 2-4? I would expect these to be absolute value or determinant symbols, but it's not clear from the text what they represent. You might see the MEE paper "Some guidance on using mathematical notation in ecology" by Andrew M. Edwards and Marie Auger-Méthé 2018 for some specific advice: <https://besjournals.onlinelibrary.wiley.com/doi/10.1111/2041-210X.13105>.

Done. Equations also updated as per suggestions. $||$ symbols in equations 2-4 represented cardinality of the sets (number of pathogens/neighbouring species).

Response to referee 2

Comments to the Author(s)

The manuscript by Wardeh et al addresses an interesting and hugely important topic, the arrival of new zoonoses in the human population. Specifically the authors tackle the task of identifying important species in the network where nodes are species. They do this by considering the networks of shared pathogens between species. After a preliminary analysis to identify suitable network metric the authors then use machine learning to identify traits that affect the species propensity to harbor potential zoonoses and also identify particular species that are likely to be transferred to humans.

This topic is of great importance and will be of interest to a very broad readership ranging from ecology and epidemiology to microbiology and medicine. The subject matter is therefore well worthy of publication Proceedings B. The authors tackle the overarching question with a novel and interesting approach. However, due to some particular choices that have been made and also issues with the write up the results are much less convincing than they could be.

We like to thank the referee for their insightful comments, key suggestions, and for the efforts they made reading our paper. Below we attempt to answer their comments

Major issues:

1) First, this paper was a tough read, both generally speaking and particular by the standards of Proceedings B. In terms of grammar and orthography the writing is fine. The overall organization into sections, where methods are separated often complicates the presentation of theory papers, and this one is no exception to this rule. However, my main problem with the text was that it is rife with numerical factoids, but lacks the context to put them into perspective. In other words it's hard to see the forest for the trees. This style is very common in experimental biology papers and also in some conceptually simple data analysis, where the basics of the approach are clear. However, in a complex study like the present one it conceals the major lines of thinking. I found myself endlessly rereading paragraphs in search for certain bits of information. Also it was very hard to connect the headline messages to the actual nuts and bolts. I come back to this point several times below.

Our response: We thank the referee for raising the readability issue. Following their suggestions we have improved the readability of our paper in number of ways.

- 1) We updated our key figures (Figure 3 and Figure 4 – attached below) to contain more results.
- 2) We shortened our results sections and supplemented all our results with 95% confidence intervals.
- 3) We added further explanations to our Indirect Influence measure as highlighted below.

These changes are integrated with our answers to the remainder of the points raised by the referee, and we have highlighted examples of these changes as part of our answer to these comments.

2) Perhaps because of the above, my central issue with this manuscript is that I am not convinced the results are informative. The manuscript tackles deep and difficult questions for which little ground truth is available. Any faith in the validity of the result must thus stem from trust in the procedures by which the results are obtained. For instance reading the main text I had no clear understanding of how the network metrics were selected, this was only a little cleared up by considering the supporting files. The authors use a combination of PCA and k-means for this task. In my assessment these are blunt outdated tools and it has been well demonstrated that they can fail spectacularly. Today there are much better methods for these tasks diffusion maps, umap that deal more gracefully with nonlinearity and high-dimensional data sets. The machine learning methods are closer than state of the art, however the methods listed are classifiers and it is not clear to me in how far the problem that is addressed here is a classification problem? This is probably related to my inability to understand the red line in the reasoning, described above.

Our response:

I) Selection of centrality measures: we have found PCA to be a useful and current tool to selecting our centrality measures. We have provided additional references to highlight use of PCA to select centrality measures in context of protein interactions networks (reference 27, listed below, 2018). The method listed in this current reference is very similar to ours. Regarding umap: Our understanding is that while umap (and similar) is a powerful dimension reduction and visualisation technique, some of

umap's known weaknesses (McInnes et al, 2018¹) are that embedding space have no specific meaning, unlike PCA where the dimensions are the directions of greatest variance in the source data. Furthermore, since umap does not have an equivalent of factor loadings that a linear technique like PCA has, we couldn't use it to select centrality measures. We wanted from the start to select representatives from the standard measures, rather than reducing the dimensionality of the centrality space (which is another option) as we wanted to keep the definition (and biological interpretation) of the selected measures intact (e.g. degree centrality, betweenness etc). We only used k-means for visualisation purposes, coupled with standard correlation (also reported). But the selection process was done via contribution to the first two dimensions of PCA (on average).

II) learning algorithms used: we selected these methods precisely because they are capable of handling both classification and regression tasks, which provided us with uniform set of underlying learners suitable for both our tasks – classification of mammalian hosts into two categories: reservoirs of zoonoses or not, and regression tasks of explaining centralities and number of zoonoses harboured by each species. We selected our methods for their accessibility to a wider audience, ease of reproduction (via the R package *caret* and *caretEnsembles* or similar packages in Python, or standard programming languages) and the good results produced. Additionally, we were able to integrate them all into single training, testing and prediction pipeline through our selected R packages.

Our changes - New reference:

27. Ashtiani M, Salehzadeh-Yazdi A, Razaghi-Moghadam Z, Hennig H, Wolkenhauer O, Mirzaie M, et al. A systematic survey of centrality measures for protein-protein interaction networks. *BMC Syst Biol.* 2018;12(1):80. <https://bmcsystbiol.biomedcentral.com/articles/10.1186/s12918-018-0598-2>

3) The manuscript tries to convince the reader using statistical arguments. Admittedly, my knowledge of statistics may be insufficient to fully appreciate the abundance performance metrics presented, but many readers will feel the same. What I am looking for is a compelling argument why I should trust

¹ McInnes L, Healy J, Melville J. UMAP: Uniform Manifold Approximation and Projection for Dimension Reduction. 2018.

the results. The information I get is that the model "explains" much of the variation in the data, but given the size of the search space this alone is not a compelling argument. My understanding is that the statics fall short of giving us actual error bounds on the results, or is this not the case?

Our response: We thank the referee for highlighting the issue of error-bounds. We have complemented our results with error bounds based on medians and empirical 95% confidence intervals to present confidence levels in all results (model results, model performance metrics and variable importance).

Changes:

Models statistics –Subsection - Ensembles Construction (lines 255- 258)

We validated our ensembles and their constituents using 10-fold cross validation. We repeated this process 100 times, to attend to uncertainties in the cross-validation processes, and to generate empirical confidence intervals. Predictions were generated using median values of these repeats.

Readability – we have shortened our results section, and made more readable. Below we list one example of these changes.

Subsection – Ensemble models to explain centrality and influence in networks of shared-pathogens (lines 276 – 286):

Neighbours' phylogenetic specificity (SND) was the top predictor overall of II (median = 26.4% - 95% CIs [9.24%, 51.1%]), and ODC (16.7% - [5.89%, 41.0%]). However, there were variations across networks (Figure 2). ODC in network of DNA viruses was best explained by species order, particularly, Carnivora (median = 11.3% - 95% CIs [10.5%, 12.7%]). ODC in networks of bacterial agents was best explained by pathogen diversity VarSTD as follows: all bacteria (median = 18.4% - 95% CIs [17.3%, 19.7%]), Gram + (21.7% - [20.7%, 23.1%]), Gram- (20.2% - 95% CIs [18.9%, 21.4%]).

Research effort was the top predictor of OBC (median = 30.2% - 95% CIs [20.5%, 69.1%]), followed by domestication status (median = 22.4% - 95% CIs [2.96%, 47.1%]) and geographical range (median = 11.1% - 95% CIs [5.09%, 21.9%]). The influence of taxonomic orders, traits, habitat, and diet predictors varied per taxa of pathogen, and centrality measure studied as highlighted in Figure 2 (Supplementary Note 4 provides full details).

4) Perhaps because of the statistical / empirical approach the manuscript is almost free of "theory" and hence theoretical points that need to be discussed remain unaddressed.

My central concern is the following: The main motivating idea that runs through the introduction is to identify reservoirs of future zoonosis for humans. It is not clear to me that this information is in principle discoverable by analysing the network of pathogens that have been shared in the past. The analysis of traits and habitat factors etc is promising, but in my reading the final step that needs to be made, essentially niche modelling, does not get across clearly.

I was actually waiting for the manuscript to make this final arc but then it took the wrong turn. The conclusions seem to highlight domesticated species as the ones we share most pathogens with. This is clearly true, but seems to forget the central question. Equally clearly, domesticated mammals are unlikely sources of undiscovered zoonoses, as we have already been in contact with these species for a long time.

Our response: We fully agree with the referee regarding the issues raised and have revised our results sections to show more tangible results. We believe our updated Figure 4 now contains more informative results. We have plotted for every taxa of pathogens, predicted number of zoonoses against total number of pathogens found in host species. Furthermore, we indicated top 20 (in all) and top 10 species (in taxa) species (by number of predicted zoonoses). We separated livestock and companion animals from main plots, and indicated predicted number of zoonoses in each of these species. By colour species by order, and adjusting size to log scale we were able to show our results in a clearer way which enabled us to free the text further.

We also updated Figure 3 with predictions performed with input set including all mammalian hosts. This figure now showcase risk (as probability between 0 and 1) of each mammalian species found to be a host of at least one species of any type of pathogen (i.e. the dataset included in all pathogens models).

Changes: We attach updated figures 3 and 4.

Figure 3 – Results of our ensemble models to predict reservoirs of zoonoses. Panel A: median variable importance (relative influence) of predictors of reservoirs of zoonoses (based on the 100 runs of each model). For the purposes of this figure the contribution of order predictors were summed. Details of contribution of all predictors of each model are presented in Supplementary Note 5. Panel B: highlights predicted median probability of host species in each order harbouring at least one zoonotic pathogen. Predictions were derived from all pathogens input set (i.e. mammalian hosts of any pathogen, N=1,560). Orders illustrated are as follows (clockwise): Artiodactyla, Carnivora, Chiroptera, Cetacea, Insectivora, Lagomorpha, Perissodactyla, Primates, Rodentia, and other mammals (all remaining orders).

Figure 4 – Results of our ensemble models to explain number of zoonoses harboured by mammalian species. Panel A: median variable importance (relative influence) of predictors included in the models over 100 runs. For the purposes of this figure, the contribution of order predictors was summed. Details of contribution of all predictors over the 100 runs of each model are presented in supplementary note 6 (supplementary figure S12). **Panel B: predicted number of zoonoses in each mammalian host species. Heat-map** illustrates logged predicted number of zoonoses in livestock, horses, and domesticated dogs and cats. **Points** represent mammalian species, coloured by their order, size = log (predicted zoonoses + 1). X-axes are predicted number of zoonoses, y-axes are detected number of zoonoses. Labels are top species by number of predicted zoonoses (n=10 in all pathogens panel, n=5 in other panels).

Specific comments:

- When I hear machine learning I am nowadays expecting that the words deep neural networks and TensorFlow follow. But hear this wasn't done, is there a reason.

Deep neural networks and tools such as TensorFlow have indeed been gaining popularity. However, the field of machine learning is vast, with many mature algorithms and tools available for experimentation. No algorithm is better than all the others for all tasks. We have selected simple to interpret, well-known algorithms to form our ensembles to: 1) explore the underlying data in a variety of ways (each algorithm represented a distinct family); 2) these algorithms are accessible and do not require specialised hardware, or access to more than normal computers; this facilitates reproducibility and extension for further work by other groups.

- Why is it actually necessary to boil down the metrics before the rest of the analysis is done? Surely the machine learning approaches can deal with multiple metrics?

The referee raises an important point. We agree that many machine learning approaches can certainly cope with correlated variables such as centrality metrics. However, in the constituent models of our ensemble approach correlation greatly affect the relevance (importance/ relative contribution) of variables. Highly correlated variables (as would have been the case if we had fed all the centrality metrics to our ML models) are in effect penalised, and their contribution to the model underestimated - the redundant variables are not assigned large importance even though they may be highly correlated with the response. We therefore needed to boil down our metrics so we can better quantify their importance to pathogen sharing with humans.

- Is there actually a message from the network plots in figure 1?

The inclusion of figure 1 was threefold: 1) to showcase differences in network structure across varied pathogen taxa (e.g. helminth vs viruses); 2) to visually highlight the different contributions of mammalian orders to the network composition per pathogen taxa by sizing each node by number of pathogens in each host species and colouring by order; and 3) to contrast (across the different taxa of pathogens compared) the sharing of pathogens across mammalian orders (e.g. protozoa and bacteria are shared between different orders more than virus or helminths).

- The "Opsahl" centralities have a free parameter alpha. How was this chosen? And, why?

Alpha was fixed at 0.5 for all Opsahl calculations to strike a balance between the weighted and unweighted versions of each centrality measure depicted by an Opsahl centrality. This is especially because we also included the weighted and unweighted versions of these centralities in our calculations.

- Why is there a need for the new indirect influence? Aren't there enough network metrics already? It seems included as an afterthought after the selection of the other metrics, is this the case? Was this necessary to produce meaningful results? Why this particular form? Is there a mechanistic reasoning rather than looking being reminiscent of entropy? It seems very odd to find this specific insertion in a paper that otherwise goes on a broad trawl of methods.

The referee raises a number of important points which we attempt to answer fully.

- There are indeed many centrality metrics in the literature, each measuring specific aspects of the node's importance. However, we believe, none of the standard measures deals specifically with shared-pathogens networks. Most the standard measures included in our analyses deal with networks where the *traffic* going through the network is of the same nature (e.g. contact networks in infection models), whereby if a node A is connected to node B and node B is connected to C then there exists a “walkable” path from A to C via B. Indirect Influence (II) is specifically designed for shared pathogen networks. It measures three aspects of the host species' ability to spread the species of pathogens infecting it indirectly through its neighbours: the total number of pathogens it can indirectly spread, the frequency (which is the sum of number of pathogens spread through all neighbours) and finally the number of neighbours indirectly reachable, as highlighted in our changes listed below.
- We kept II separate to other metrics in the manuscript for two reasons: 1) Traditional measures are still very insightful to the nature of the network and many aspects of pathogen sharing (e.g. betweenness centrality). We wanted to provide means to selecting from amongst these measures, and examine which mammalian traits, phylogeny and other predictors influence these traditional metrics. 2) We wanted to show that our measure complements the above metrics. As each family of metrics examines particular aspect of node importance we wanted to show which metrics our

measure correlates with – it is similar in nature to the traditional closeness measure but it designed for shared-pathogen networks.

- The motivation of using entropy as basis for II is to capture uncertainties in pathogens destination, as a function of the original host. We have highlighted this uncertainty in the main text as listed below.

Changes

Subsection: Novel metric of node Influence – Indirect Influence (II) (lines 132- 155):

Unipartite projection of host-pathogen bipartite networks results in inevitable loss of information (39).

Let us assume, for instance, that we have three mammalian species: A, B and C, and that A and B share 10 protozoan agents, whereas B and C share 5 protozoan agents. Here we have two potential scenarios: 1) A and C do not share any protozoan pathogens, which in turns means that there is no flow of protozoan pathogens from A to C despite a path existing between the two (via B); 2) A and C share some protozoan agents (e.g. 2), this could mean that all or some or none of these pathogens are also shared with B. Both these scenarios have implications on the traditional centrality metrics analysed in the previous subsection. To address these issues we developed a novel entropy-based metric, which we term Indirect Influence (II). II captures the influence each host species exercises within the unipartite network by measuring the number and frequency of pathogens this host spreads indirectly through its neighbouring species if it were to interact with each of them in isolation to the remainder of the network. Entropy enables us to capture uncertainties in the destination of pathogens as a function of the original host (40) at three levels - i) a species which share a few pathogens with many neighbours: entropy captures uncertainty in which neighbour it could influence indirectly; ii) a species which shares many pathogens with few neighbours: entropy allows us to capture uncertainty in pathogens shared; iii) a species which share many pathogens with many neighbours will have high centrality due to uncertainties in both pathogens shared and neighbours influenced. In addition, using entropy allowed us to avoid assessing the many paths connecting all node pairs and instead, focus on the potential of the node (i.e. the host species) to diversify pathogen propagation (40,41).

References:

40. Nikolaev AG, Razib R, Kucheriya A. On efficient use of entropy centrality for social network analysis and community detection. *Soc Networks* [Internet]. 2015; 40:154–62.
41. Qiao T, Shan W, Zhou C, Qiao T, Shan W, Zhou C. How to Identify the Most Powerful Node in Complex Networks? A Novel Entropy Centrality Approach. *Entropy*. 2017;19(11):614.
<http://www.mdpi.com/1099-4300/19/11/614>

Let me finish by admitting that I have struggled with this manuscript quite a bit. Please accept my apologies for being very slow with review and likely misinterpreting some important points.

We thank the referee for their effort. As stated above, we have attempted to improve the readability of our paper by taking their comments and suggestions on board, improving our results section and enriching our figures as highlighted above.

Appendix B

Response to referee 1

Comments to the Author(s).

The manuscript by Wardeh et al. presents a data-driven approach to explain the structure of mammalian shared pathogen networks, which includes quantitative comparisons of various machine learning algorithms, a comparison of network statistics that characterize structure, and new network metrics. The authors have done a great job addressing my comments from the original submission.

The revised manuscript is more clear, and I have no additional major comments.

We thank referee 1 for reading, and re-reading our paper. Their insight and comments have been essential and help it become a much better paper.

Minor suggestions

In figure 4, a dashed line along the diagonal for the scatter plots might be better than the smooth fits that are currently presented, which seem to be sensitive to outliers (e.g., the Gram - panel). This is a very minor (and mostly aesthetic) comment, however.

Our response: we have updated Figure 4 with the above suggestions, as highlighted below.

Figure 4 – Results of our ensemble models to explain number of zoonoses harboured by mammalian species.

Panel A: median variable importance (relative influence) of predictors included in the models over 100 runs. For the purposes of this figure, the contribution of order predictors was summed. Details of contribution of all predictors over the 100 runs of each model are presented in supplementary note 6 (supplementary figure S12).

Panel B: predicted number of zoonoses in each mammalian host species. Heat-map illustrates logged predicted number of zoonoses in livestock, horses, and domesticated dogs and cats. Points represent mammalian species, coloured by their order, size = log (predicted zoonoses + 1). X-axes are predicted number of zoonoses, y-axes are detected number of zoonoses. Labels are top species by number of predicted zoonoses (n=10 in all pathogens panel, n=5 in other panels).

Response to referee 2

Comments to the Author(s).

The revision has addressed my concerns to some extent. I am still not 100% confident in the results, but this is unlikely to improve with further revisions. The manuscript takes an interesting approach to an interesting topic and is not misleading. I am therefore in favour of publication.

We are grateful to referee 2 for their comments, insights, and critique of our paper, which helped immensely. We understand their concerns and hope they will help us when planning and writing future work.